**Exploring the variations in ambient BTEX in urban Europe and its environmental health implications**

Xiansheng Liu,[1] Xun Zhang,[2*] Marvin Dufresne,[3] Tao Wang,[4] Lijie Wu,[2] Rosa Lara,[1] Roger Seco,[1] Marta Monge,[1] Ana Maria Yáñez-Serrano,[1,5,6] Marie Gohy,[7] Paul Petit,[7] Audrey Chevalier,[8] Marie-Pierre Vagnot,[9] Yann Fortier,[9] Alexia Baudic,[10] Véronique Ghersi,[10] Grégory Gille,[11] Ludovic Lanzi,[11] Valérie Gros,[12] Leïla Simon,[12] Heidi Hellen,[13] Stefan Reimann,[14] Zoé Le Bras,[14] Michelle Jessy Müller,[14] David Beddows,[15] Siqi Hou,[15] Zongbo Shi,[15] Roy M Harrison,[15] William Bloss,[15] James Dernie,[16] Stéphane Sauvage,[3] Philip K. Hopke,[17,18] Xiaoli Duan,[19] Taicheng An,[20] Alastair Lewis,[21] Jim Hopkins,[21] Eleni Liakakou,[22] Nikolaos Mihalopoulos, [22,23] Xiaohu Zhang,[24] Andrés Alastuey,[1] Xavier Querol,[1] Thérèse Salameh[3*]

[1]Institute of Environmental Assessment and Water Research (IDAEA-CSIC), 08034 Barcelona, Spain

[2]Beijing Key Laboratory of Big Data Technology for Food Safety, School of Computer and Artificial Intelligence, Beijing Technology and Business University, Beijing 100048, China

[3]IMT Nord Europe, Institut Mines-Télécom, Univ. Lille, Centre for Energy and Environment, F-59000 Lille, France.

[4]Shanghai Key Laboratory of Atmospheric Particle Pollution and Prevention, Department of Environmental Science and Engineering, Fudan University, Shanghai, 200433, China

[5]CREAF, E08193 Bellaterra (Cerdanyola del Vallès), Catalonia, Spain

[6]CSIC, Global Ecology Unit, CREAF-CSIC-UAB, E08193 Bellaterra (Cerdanyola del Vallès), Catalonia, Spain

[7]Institut Scientifique de Service Public (ISSeP), 4000 Liège, Belgium

[8]Atmo Grand-Est (AtmoGE), 67300 Schiltigheim, France

[9]Atmo Auvergne-Rhône-Alpes (AtmoAURA), 69500 Bron, France

[10]Airparif, Air Quality Monitoring Network for the Greater Paris area, 7 rue Crillon, 75004 Paris, France

[11]AtmoSud, 13006 Marseille, France

[12]Laboratoire des Sciences du Climat et de l'Environnement (CEA-CNRS-UVSQ, IPSL), CAE/Orme des Merisiers, 91191 Gif sur Yvette, France

[13]Finish Meteorological Institute (FMI), FI-00560 Helsinki, Finland

[14]Swiss Federal Laboratories for Materials Science and Technology (Empa), 8600 Dübendorf, Switzerland

[15]School of Geography Earth and Environmental Sciences, University of Birmingham, B15 2TT Birmingham, United Kingdom

[16]Ricardo, W2 6LA London, United Kingdom

[17]Institute for a Sustainable Environment, Clarkson University, Potsdam, NY 13699 USA

[18]Department of Public Health Sciences, University of Rochester School of Medicine and Dentistry, Rochester, NY 14642 USA

[19]sSchool of Energy and Environmental Engineering, University of Science and Technology Beijing, Beijing, China

[20]School of Environmental Science and Engineering, Guangdong University of Technology, Guangzhou 510006, China

[21]Wolfson Atmospheric Chemistry Laboratories, Department of Chemistry, University of York, Heslington, York, YO10 5DD, UK

[22]Institute for Environmental Research and Sustainable Development, National Observatory of Athens, 15236, Athens, Greece

[23]Environmental Chemical Processes Laboratory, Department of Chemistry, University of Crete, 70013, Crete, Greece

[24]National Engineering and Technology Center for Information Agriculture, Nanjing Agricultural

University, Nanjing 210095, China.

*Correspondence to: Xun Zhang(zhangxun@btbu.edu.cn); Thérèse Salameh (therese.salameh@imt-nord-europe.fr)*

**Abstract**

BTEX (benzene, toluene, ethylbenzene, m,p,o-xylene) are significant urban air pollutants. This study examines BTEX variability across seven European countries using data from 22 monitoring sites in different urban settings (urban background, traffic, industry, and suburban background). Results indicate that the relative abundance of BTEX in urban areas follows the order: toluene > benzene > m,p-xylene > o-xylene > ethylbenzene, with median mixing ratios of $266 \pm 152$, $163 \pm 74$, $129 \pm 88$, $53 \pm 35$, and $45$

$\pm 27$ ppt during the years 2017-2022, respectively. Seasonal trends show benzene had similar median concentrations across urban background, traffic, and industrial sites, indicating mixed sources. Toluene levels were highest in traffic and industrial areas, highlighting road traffic and industrial emissions. Ethylbenzene and xylenes showed equivalent levels in traffic and industrial areas but were lower in urban backgrounds. Peak BTEX levels occurred during morning and evening rush hours, linked to traffic,

heating, and atmospheric stagnation. B/T ratios ranged from $0.29\pm0.11$ to $1.35\pm0.95$, and X/E ratios ranged from $1.75\pm0.91$ to $3.68\pm0.30$, indicating primary pollution from local traffic, followed by solvents, coatings, and biomass burning. Lifetime Cancer Risk for BTEX exposure were below the definite risk threshold ($10^{-4}$) but above the permissible risk level ($10^{-6}$), suggesting moderate risk from benzene and ethylbenzene, particularly in traffic and industrial areas. Additionally, the health index of BTEX at

monitoring sites were generally lower than the threshold limit value, suggesting a low non-carcinogenic risk overall. This study offers essential insights into BTEX pollution in European urban environments.

**1 Introduction**

Atmospheric volatile organic compounds (VOCs) are significant precursors of tropospheric ozone

($O_3$) (Grosjean and Seinfeld, 1989) and secondary organic aerosols (SOA) (Derwent et al., 2010). Among the most prevalent VOCs, BTEX—comprising benzene, toluene, ethylbenzene, and xylenes—is a typical component of air pollution (Miri et al., 2016). Due to their reactivity with hydroxyl radicals (·OH) during daylight and nitrate radicals (NO$_3$·) both day and night, they can generate additional radicals, such as peroxyalkyl and hydroperoxy radicals, which further oxidize nitric oxide (NO) to nitrogen dioxide (NO$_2$)

(Garg and Gupta, 2019; Ghaffari et al., 2021). This process contributes to increased concentrations of

tropospheric $O_3$ and secondary organic aerosol (SOA) formation (Ng et al., 2007). Additionally, exposure to BTEX is heightened by solvent evaporation, vehicular traffic, and emissions from fossil fuel extraction, which are increasingly occurring near densely populated areas (Bolden et al., 2015; Salameh et al., 2019; AQEG-EU, 2020; Liu et al., 2023). In addition, the removal of toxic octane enhancers like lead and methyl tertiary-butyl ether (MTBE) from gasoline allowed refiners to increase the volume of aromatics in gasoline to meet anti-knock requirements and enhance octane ratings (Yang et al., 2019). Thus, due to its utility, ubiquity, and economic importance, BTEX is likely to remain a persistent environmental pollutant in the air for the foreseeable future (Bolden et al., 2015; Davidson et al., 2021).

Furthermore, there is a body of evidence that BTEX has the potential to irritate various organs within the human body, including the respiratory tract, lungs, bronchi, skin, and heart, and that exposure to elevated concentrations can result in acute effects such as dizziness and vomiting (Davidson et al., 2021; Li et al., 2021; Ogbodo et al., 2022). According to the International Agency for Research on Cancer (IARC), benzene is recognized as a significant public health threat substance and is classified as a carcinogen (WHO, 2016). Additionally, the US EPA's carcinogenicity classification system categorizes benzene as a human carcinogen (Group A), supported by ample evidence of its carcinogenic effects in humans. On the other hand, toluene, ethylbenzene, and xylenes are not classified as to human carcinogenicity (Group D) with inadequate (or no) evidence, while they have important non-carcinogenic toxic potentials (Durmusoglu et al., 2010). In 1990, the U.S. Clean Air Act Amendments classified seven categories of VOCs (https://www.epa.gov/haps), including BTEX and other three VOCs (n-hexane, 1,3-butadiene, and styrene) as hazardous pollutants, highlighting their potential to cause human cancer and other serious health issues. Simultaneously, the Environmental Protection Directory issued by the Ministry of Ecology and Environment of the People's Republic of China designates BTEX and styrene as high-pollution and high-risk environmental pollutants (Song et al., 2018). Furthermore, the European Air Quality (DIRECTIVE (EU), 2024) establishes the latest limit value of 3.4 $\mu g/m^3$ for the annual mean of benzene in ambient air, with all member countries required to monitor and report on this carcinogenic compound to the European Environment Agency (EEA) database (https://www.eea.europa.eu/en). Since BTEX members are small molecules with lipophilic characteristics, they easily penetrate the human body (Zahed et al., 2024). However, studies focusing on BTEX within the European Union (EU) have primarily centered on Member States. For instance, Borbon et al. (2018) conducted long-term observations of hydrocarbons, including TEX, in traffic and urban background locations in London, Paris, and Strasbourg. They estimated the relative importance of traffic emissions for TEX in each city and found that traffic emissions would no longer dominate TEX concentrations in urban areas of Europe. Despite this, studies addressing the overall BTEX pollution status and health risk assessment at the EU level remain relatively insufficient.

RI-URBANS (Research Infrastructures Services Reinforcing Air Quality Monitoring Capacities in European Urban and Industrial Areas, funded by the EU's Horizon 2020 research and innovation program, 101036245) is a European research project that demonstrates the application of advanced air quality service tools in urban Europe to improve the assessment of air quality policies, including a more accurate evaluation of health effects. In this context, this study focused on collecting and evaluating both online and offline BTEX data from 19 cities (total 22 monitoring sites) across seven European countries

(Belgium, Finland, France, Greece, Switzerland, Spain, and the United Kingdom). The study's objectives included: (i) comparing BTEX concentrations across Europe; (ii) identifying sources of BTEX; and (iii) assessing the health impacts of BTEX. By comprehensively assessing BTEX levels in major European cities and their harmful effects on human health, this study aims to provide data support for the coordinated management of BTEX monitoring across Europe.

## 2 Methodology

### 2.1 Instrumentation

The instrumentation used for measuring BTEX at different stations, as described in Table 1, encompassed 3 industrial (IND) sites, 2 traffic (TR) sites, 16 urban background (UB) sites, and 1 suburban background (SUB) site. Briefly, in this study the BTEX was measured by Thermal Desorption Gas Chromatography with Flame Ionization Detectors (TD-GC-FID/2FID), Thermal Desorption Gas Chromatography-Mass Spectrometry (TD-GC-MS), Proton Transfer Reaction-Time of Flight-Mass Spectrometry (PTR-TOF-MS), PTR-Quad-MS, and passive samplers. It should be noted that comparing such heterogeneous datasets, without an intercomparison exercise, is a limitation of the study.

In addition, as shown in Table 1, the monitoring periods differed for each site. We considered only BTEX data obtained from urban sites between the years 2017 and 2022, with the exception of HEL_UB (Helsinki) (only February 2016) and ATH _UB (Athens) (only 2016-2017) that provided shorter timeseries, to reduce the uncertainties when comparing BTEX mixing ratios and health risk assessment between sites due to long term trends at each site (Table 1, Figure 1). Also, BCN_UB (Barcelona) and PAR_SUB (Paris-SIRTA) monitored the combined levels of ethylbenzene and xylenes. However, because the PTR-MS technique used at these sites does not distinguish between isomers (Simon et al., 2023), we excluded these data from our analysis, when comparing BTEX mixing ratios. Additionally, BAQS_UB (Birmingham) did not detect m,p-xylene.

*Table 1. Details of measuring sites and instrumentation used for offline and online BTEX datasets in this study (IND, Industry; TR, Traffic; UB, Urban background; SUB, Suburban background).*

| City (Country) | Acronym | Instrument type/model | Start date | End date | Types |
|---|---|---|---|---|---|
| Athens (GR) | ATH_UB | TD-GC-FID | 01/03/2016 | 28/02/2017 | online |
| Angleur (BE) | ANG_UB | Passive samplers | 01/01/2018 | 31/12/2021 | offline |
| Birmingham (UK) | BAQS_UB | TD-GC-FID | 14/04/2021 | 06/06/2022 | online |
| Barcelona (ES) | BCN_UB | PTR-TOF-MS | 07/10/2021 | 31/12/2022 | online |
| Charleroi (BE) | CHM_UB | Passive samplers | 02/01/2010 | 30/12/2021 | offline |
| Grenoble (FR) | GRE_UB | TD-GC-2FID Perkin-Elmer | 16/01/2015 | 29/12/2022 | offline |
| Helsinki (FI) | HEL_UB | TD (Markes) -GC-MS (Agilent) TD-GC-MS Perkin-Elmer | 20/01/2011 28/01/2016 | 18/11/2011 24/02/2016 | online online |
| Herstal (BE) | HET_UB | Passive samplers | 01/01/2013 | 31/12/2021 | offline |
| Lodelinsart (BE) | LDS_UB | Passive samplers | 02/01/2018 | 30/12/2021 | offline |
| Eltham (UK) | LND_UB | TD-GC-FID | 01/01/2008 | 01/01/2022 | online |

| | | | | | |
|---|---|---|---|---|---|
| Marseille (FR) | MAR_UB | TD-GC-2FID Perkin-Elmer | 31/01/2019 | 13/08/2020 | online |
| Mons (BE) | MON_UB | Passive samplers | 02/01/2010 | 30/12/2021 | offline |
| Namur (BE) | NAM_UB | Passive samplers | 01/01/2018 | 31/12/2021 | offline |
| Paris - Paris 1er Les Halles (FR) | PAR_UB | TD-GC-FID (30min) | 01/01/2020 | 01/01/2022 | online |
| Strasbourg Ouest (FR) | STB_UB | TD-GC-FID | 08/07/2002 | 30/08/2021 | online |
| Zurich (CH) | ZUR_UB | TD-GC-2FID | 01/01/2016 | 31/12/2017 | online |
| Helsinki (FI) | HEL_TR | TD-GC-MS Perkin-Elmer | 21/08/2019 | 11/09/2019 | online |
| London (UK) | LND_TR | TD-GC-FID | 01/01/2007 | 01/01/2022 | online |
| Lyon - Feyzin stade (FR) | LYO1_IND | TD-GC-2FID Perkin-Elmer | 20/10/2004 | 10/02/2023 | online |
| Lyon - Vernaison (FR) | LYO2_IND | TD-GC-2FID Perkin-Elmer | 30/03/2009 | 10/02/2023 | online |
| Mouscron (BE) | MSR_IND | Passive samplers | 02/01/2010 | 30/12/2021 | offline |
| Paris-SIRTA (FR) | PAR_SUB | PTR-Quad-MS Ionicon | 18/01/2020 | 30/12/2021 | online |

**2.2 Risk assessment**

Health risk assessments were conducted to evaluate the impacts on human health from exposure to atmospheric pollutants through inhalation, ingestion, and dermal penetration (Li et al., 2014; Yao et al., 2019; Nie et al., 2020; Zahed et al., 2024). Among the aforementioned pathways, inhalation is regarded as the main route. Therefore, in this study, the health risks associated with inhalation exposure were assessed. The Lifetime Cancer Risk (LCR) were performed based on the method proposed by the U.S. Environmental Protection Agency (EPA) in 2009 (EPA-540-R-070-002) (EPA, 2009). This method was employed to evaluate carcinogenic risks of inhaled benzene and ethylbenzene on the population. The formula for calculation is as follows: $LCR = CA \times IUR$, where CA represents the concentration of pollutants in the atmosphere ($\mu g/m^3$). The mixing ratio was converted into concentrations for benzene and ethylbenzene. The Inhalation Unit Risk (IUR) is the unit risk value for inhalation exposure ($\mu g/m^3$)$^{-1}$, indicating the maximum probability of developing cancer when exposed to a certain dose of a pollutant through inhalation. The IUR values for benzene and ethylbenzene are $7.8 \times 10^{-6}$ and $2.5 \times 10^{-6}$ ($\mu g/m^3$)$^{-1}$, respectively, as derived from the U.S. EPA IRIS (EPA, 2020).

According to U.S. EPA standards, LCR represents the probability of cancer occurrence in the exposed population, usually expressed as the ratio of the number of cancer cases per unit population. If the LCR falls below $10^{-6}$ (indicating an increase of 1 cancer case per 1 million people in a lifetime), it is widely considered as negligible. Some jurisdictions consider risks as high as $10^{-4}$ as tolerable, and those higher than this as requiring urgent action. Specially, in this study, we adopted U.S. EPA values as they provide a standardized framework specifically designed for environmental exposures (Phillips and Moya, 2013). The EPA's health risk assessment values (LCR and RfC) are widely utilized due to their systematic approach and extensive data basis for chronic exposure scenarios. While the EU's Occupational Exposure Limits (OEL) are established for workplace environments, they primarily address acute exposure risks in occupational settings (Högberg and Järnberg, 2023). Therefore, they are not entirely suitable for assessing chronic, low-level exposure risks among the general public.

Additionally, the non-cancer health risks of human exposure to all identified BTEX compounds at the monitoring sites were assessed. The results are presented as the hazard index (HI), defined as the ratio of long-term intake to the reference dose for respiratory exposure. According to U.S. EPA standards, an HI value equal to or greater than 1 indicates potential adverse health effects for the exposed population under current environmental conditions. The HI is calculated using the formula: $HI = CA \times 1/RfC$, where RfC represents the reference concentration of VOC species. The RfC values for BTEX compounds are 30, 5000, 1000, and 100 (μg/m³), respectively, as determined by the U.S. EPA (EPA, 2020).

**2.3 Data Treatment**

The data quality was assessed and evaluated by two ACTRIS CiGas (https://www.actris.eu/topical-centre/cigas) units at IMT Nord Europe - France and EMPA-Switzerland, based on ACTRIS recommendations and guidelines (Laj et al., 2024). Some sites indicated that they are following the ACTRIS guidelines especially for GC measurements (https://www.actris.eu/sites/default/files/inline-files/WP3_D3.17_M42_0.pdf). However, the data evaluation showed that some sites had outliers, which were deleted when analyzed. Meanwhile, the International Union of Pure and Applied Chemistry strongly recommends using mixing ratios (Schwartz and Warneck, 1995). Therefore, in this study, BTEX levels are presented as median mixing ratios (MED) ± median absolute deviation (MAD) in parts per trillion by volume (ppt) to reflect the non-normal distribution of the data. Statistically significant differences in BTEX mixing ratios at various monitoring sites and during different seasons were assessed using the Mann-Whitney U test for pairwise comparisons and the Kruskal-Wallis test for overall differences in medians (Kruskal and Wallis, 1952). These statistical analyses were performed using the SPSS Software (IBM SPSS Statistics 25, Chicago, IL, USA).

**3 Results and discussion**

**3.1 Status of BTEX data availability and mixing ratios in urban Europe**

The results showed that, in urban areas, the relative abundance of BTEX followed this order: toluene > benzene > m,p-xylene> o-xylene > ethylbenzene with median mixing ratios of 266 ± 152, 163 ± 74, 129 ± 88, 53 ± 35, and 45 ± 27 ppt during the years 2017-2022, respectively (Figure 1). Comparing different types of locations, the average/median mixing ratios of BTEX were higher at industry (IND) and traffic (TR) sites, while lower levels were found at urban background (UB) sites (Figure S1). These observed differences in BTEX concentrations suggest potential contributions from transportation and industrial activities in the studied urban areas. Therefore, to strengthen our conclusions, we applied urban enhancement ratios (ER) (Salameh et al., 2019), estimating the slopes of least-square linear regressions between each TEX compound and benzene. By using ER, we reduce the sensitivity of the analysis to background conditions, dilution, and air-mass mixing compared to using absolute concentrations (Salameh et al., 2019). Our results show spatial differences in the ER values for TEX/B, with the highest ratios observed at TR sites, followed by UB sites, and the lowest at IND sites. Specifically, the slopes were 2.09±0.05 for T/B, 0.37±0.01 for E/B, 1.21±0.03 for m,p-X/B, and 0.48±0.01 for o-X/B at TR sites, 1.57±0.02 for T/B, 0.23±0.00 for E/B, 0.71±0.01 for m,p-X/B, and 0.27±0.00 for o-X/B at UB sites, and 0.37±0.01 for T/B, 0.13±0.00 for E/B, 0.29±0.01 for m,p-X/B, and 0.13±0.00 for o-X/B at IND sites. A

similar trend was observed in the seasonal variations, with ER values generally following the order TR > UB > IND. Notably, for UB and TR sites, the ER for TEX/B was higher in summer, while for IND sites, the ER was lowest during summer (Table S1). These findings suggest that the additional evaporative sources, potentially related to traffic or solvent usage, particularly at urban background sites, may contribute to the observed seasonal and spatial variations.

**3.2 Spatial variations of BTEX mixing ratios**

Spatial variation in BTEX mixing ratios across urban Europe is significant. For instance, ATH_UB (Athens) had the highest median BTEX levels (2768 ± 4117 ppt) among all monitoring sites, followed by BCN_UB (Barcelona) (622 ± 312 ppt). This is primarily because these two sites are located in the eastern part of the Mediterranean basin, where the combination of a temperate climate (mild and rainy winters versus hot and dry summers) favors the development of severe air pollution episodes (Monks et al., 2009; Im and Kanakidou, 2012). Nonetheless, compared to the period before 2000, the levels of benzene and other BTEX compounds have shown a decreasing trend due to the successful implementation of air quality measures in Greece, such as the extension of metro lines and the use of catalytic converters in cars (Panopoulou et al., 2021). For instance, benzene levels at traffic sites have decreased significantly, dropping by as much as eightfold, from approximately 12520 ppt in 1994 to about 1565 ppt in 2016 (Panopoulou et al., 2021). Similarly, at urban monitoring stations, benzene concentrations fell sharply, from around 4695 ppt during the period of 1993–1996 to between 313 and 1565 ppt in 2016 (Panopoulou et al., 2021).

However, compared to other European cities, pollution in Athens remains relatively severe, largely due to traffic but especially to wood burning impact for heating (Panopoulou et al., 2018). For more details on the spatiotemporal variation and source apportionment of BTEX and other VOCs at this site, refer to (Panopoulou et al., 2021).

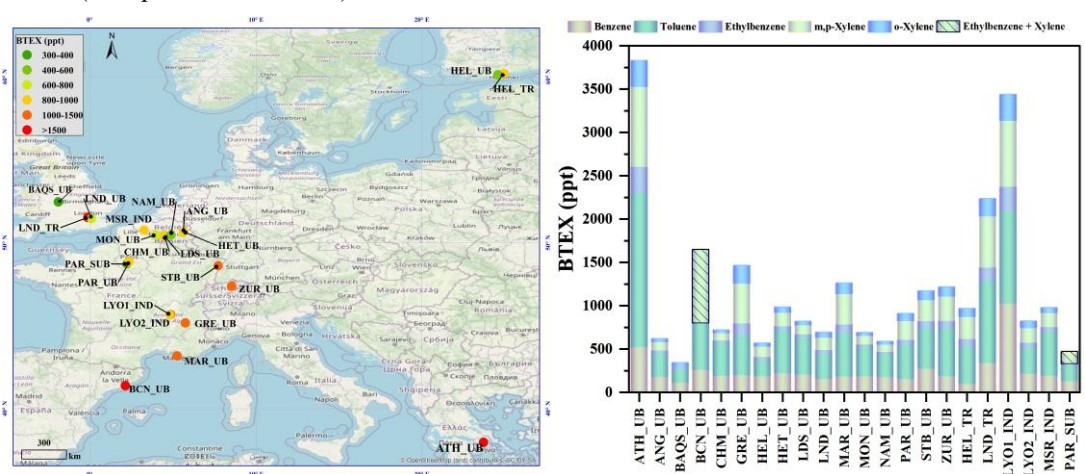

Figure 1. Average mixing ratios of BTEX at 22 European sites (left panel) and the contribution of BTEX at each monitoring site (right panel) during 2017-2022 (but HEL_UB only during February 2016 and ATH_UB only 2016-2017). Please note that the BTEX observations at BCN_UB and PAR_SUB were not able to distinguish the mixing ratios of ethylbenzene separate from xylenes, and at BAQS_UB did not include m,p-xylene data. © OpenStreetMap contributors. Distributed under a Creative Commons BY-SA License.

**3.3 Temporal variations of BTEX mixing ratios**

Figure S2 depicts the temporal variations of BTEX mixing ratios at the different monitoring stations. The mixing ratios curves of BTEX species exhibited irregular fluctuations over the whole sampling period, but the mixing ratios of each component showed highly similar temporal variations. These results indicated a significant correlation among their concentrations, and consequently suggested similar or identical sources for the BTEX species (Table S2). The highest correlation was observed between o-xylene and m,p-xylene at all monitoring stations ($r^2$=0.62-0.99), suggesting that they had similar emission sources (*e.g.*, paints, finishing, fuels, and solvent). Conversely, the correlation between benzene and xylene exhibited some variability across different sites. In the majority of locations, the correlation falls within the range of $r^2$=0.30 to 0.59. However, lower correlations ($r^2$<0.20) were observed at BAQS_UB (Birmingham) and LYO1_IND (Lyon-Feyzin stade), while stronger and significant correlations ($r^2$>0.70) were noted at ATH_UB (Athens), HEL_TR (Helsinki) and LDS_UB (Lodelinsart). This variability may be attributed to their respective emission sources, atmospheric lifetimes, and differences in their photo-reactivity within distinct environment (Monod et al., 2001). Indeed, benzene has a rate constant (kOH) for reaction with hydroxyl radicals (·OH) of $1.2 \times 10^{-12}$ cm³ molecule$^{-1}$ s$^{-1}$ and an atmospheric lifetime of about 9.5 days. In contrast, xylene has a significantly higher kOH, ranging from $14 \times 10^{-12}$ cm³ molecule$^{-1}$ s$^{-1}$ for o-xylene to $23.1 \times 10^{-12}$ cm³ molecule$^{-1}$ s$^{-1}$ for m-xylene, with a much shorter atmospheric lifetime between 10 and 20 hours (Atkinson and Arey, 2003; Liu et al., 2023). These differing properties suggest that both photo-reactivity and source emissions crucially influence atmospheric behavior and concentrations.

3.3.1 Seasonal variation

A comparative analysis of seasonal variations (Spring, Mar-May; Summer, Jun-Aug; Autumn, Sep-Nov; Winter, Dec-Feb) in BTEX mixing ratios across different types (urban background (UB), traffic (TR), and industry (IND)) of 21 urban sites was also performed (Figure 2). The results indicate, that benzene reached very similar median mixing ratios at UB, TR and IND in all seasons, pointing to a mix of source contributions. While for toluene, the average and median mixing ratios followed this trend: TR≥IND>UB for all seasons, further indicating a relatively greater influence of traffic sources on this species (Baudic et al., 2016; Salameh et al., 2019). For ethylbenzene and xylene (EX), averaged mixing ratios measured seem to be equivalent for TR and IND (and significantly lower at UB sites).

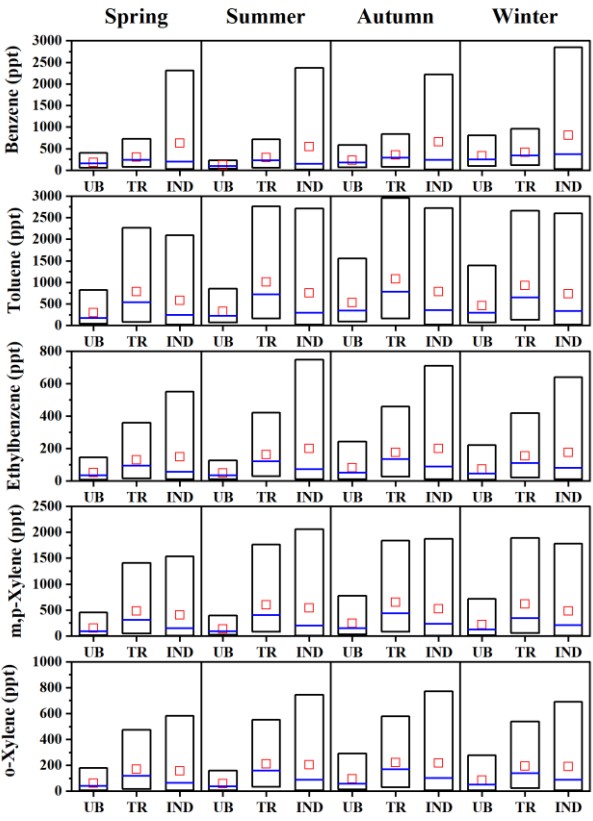

Figure 2. Seasonal variations of BTEX (benzene, toluene, ethylbenzene, xylene) mixing ratios for the different types sites (UB, urban background; TR, traffic; IND, industry). The box represents the 5th–95th percentiles of ratios. The middle line and middle square represent the median and mean values of ratios, respectively.

Figure S3 shows the BTEX mixing ratios by season for each site. Overall, BTEX mixing ratios at all monitoring sites showed substantial seasonal differences using Mann-Whitney test Mann-Whitney U test and the Kruskal-Wallis test ($p$=0.05). The extent to which there were statistically significant differences in each BTEX concentration between seasons varied slightly across different monitoring sites (Figure S3). Specifically, except for sites BAQS_UB (Birmingham), HEL_TR (Helsinki), and LYO1_IND (Lyon - Feyzin stade), the BTEX mixing ratios were lower during spring and summer and higher during autumn and winter. This finding can be attributed to seasonal differences in photochemical reactions, atmospheric conditions, and the intensity of different emission sources. When analyzing meteorological data from selected monitoring sites (Figure S4), it becomes evident that winter exhibits lower temperatures and mixing layer height, unfavorable for pollutant dispersion. Conversely, summer features more conducive diffusion conditions, likely resulting in lower pollutant concentrations. Meanwhile, the increase in BTEX mixing ratios in winter may be influenced by the emissions from residential heating and road-traffic emissions (Boynard et al., 2014; Panopoulou et al., 2018), and the lower spring and summer concentrations might be also linked to a high photochemical oxidation (Hui et al., 2019). At BAQS_UB (Birmingham), benzene was lower in summer and higher in winter, while TEX (toluene, ethylbenzene, and xylene) did not vary significantly, suggesting that in combination urban gasoline and solvent emission rates have been stabilized (AQEG-EU, 2020). Due to only summer and

autumn data being available, HEL_TR (Helsinki) exhibited higher mixing ratios for BTEX in summer compared to autumn, which can be attributed to increased evaporative emissions during the summer months. For LYO1_IND (Lyon-Feyzin stade), benzene levels were lower in summer and higher in winter, while TEX levels were lower in spring and higher in summer. This indicates that during the summer, TEX mixing ratios can mainly be influenced by additional pollution sources, such as the volatilization of paints, finishing products, solvents, and fuel evaporation.

Additionally, seasonal variations in gasoline formulations could also play a significant role. For example, increasing ethanol content in gasoline can reduce emissions of BTEX (Yao et al., 2011; Karavalakis et al., 2014). The most common formulation in Europe is E5 (containing 5% ethanol), with ethanol concentrations generally higher in the summer than in the winter (Dunmore et al., 2016).

3.3.2 Diel Variation

Only online monitoring sites with hourly resolution BTEX data were considered (ATH_UB (Athens), BAQS_UB (Birmingham), BCN_UB (Barcelona), HEL_UB (Helsinki), LND_UB (London), MAR_UB (Marseille), PAR_UB (Paris-Paris 1er Les Halles), STB_UB (Strasbourg Ouest), ZUR_UB (Zurich), HEL_TR (Helsinki), LND_TR (London), LYO1_IND (Lyon-Feyzin stade), LYO2_IND (Lyon-Vernaison), and PAR_SUB (Paris-SIRTA)). Diel variations of BTEX (Figure 3) indicated that at some sites, such as ATH_UB (Athens), BCN_UB (Barcelona), LND_UB (London), MAR_UB (Marseille), HEL_TR (Helsinki), LND_TR (London), and LYO1_IND (Lyon-Feyzin stade), double peaks varying in intensity, amplitude, and time of the maxima. Generally, the concentration of aromatic compounds is higher at night than during the day. The chemical removal process of BTEX mainly involves photochemical reactions with OH and $NO_3$ radicals (Atkinson and Arey, 2003; Carter, 2007). ·OH radicals are mainly present during the daytime, and their reaction rates with VOCs are faster than the $NO_3$ radicals present at night (Zou et al., 2015). Additionally, the higher mixing layer height during the day aids in diluting pollutants, while nighttime emissions from activities like firewood for leisure and winter heating result in higher BTEX concentrations at night compared to daytime levels (Wu et al., 2016; Liu et al., 2020).

The first of the two majors diel BTEX peaks coincides with the morning road-traffic rush hours. Particularly, BTEX mixing ratios in ATH_UB (Athens) and LYO1_IND (Lyon-Feyzin stade) are significantly ($p = 0.001$) higher than at the other sites, which may be closely related to industrial sources, off-road transport, types of vehicles, traffic flow, and other combustion-related sources. Similarly, influenced by traffic, the peaks of BTEX in LND_TR (London) were also more pronounced. After 17:00, when people mainly leave from work, traffic increases, and the mixing layer height decreases. A similar phenomenon was observed in HEL_TR (Helsinki), although this site only provided data for one month in 2019 (Table 1). Consequently, an evening peak in BTEX levels is regularly observed between approximately 18:00-19:00 at these sites (Figure 3). However, LYO1_IND (Lyon-Feyzin stade) is hardly affected by the evening rush hour. This is primarily due to the site's exposure to industrial processes, as well as wind speed and direction, particularly when the monitoring site is downwind of the industrial area. The highest levels are observed at 23:00. These phenomena further indicate that BTEX are largely associated with anthropogenic activities and specific meteorological factors. Furthermore, in addition to

traffic emissions, evaporative sources contribute significantly to the diel patterns observed in some of the BTEX. Given that evaporative emissions typically rise in the afternoon as temperatures increase (Nguyen et al., 2009), any elevation in TEX mixing ratios may be linked to their evaporative sources (Yurdakul et al., 2018). It is striking how similar the levels and variability of toluene are in BAQS_UB (Birmingham), HEL_UB (Helsinki), and PAR_SUB (Paris-SIRTA), likely reflecting similar traffic influences. In contrast, benzene levels show more variation, indicating regional influences. Focusing on Paris, similar to the findings of (Simon et al., 2023), we observe comparable levels and variability of benzene between PAR_UB and PAR_SUB (Paris-SIRTA), suggesting regional influence. However, for toluene, levels are higher in PAR_UB (Paris - Paris 1$^{er}$ Les Halles) compared to PAR_SUB (Paris-SIRTA), indicating more local traffic influences in Paris.

Similarly, the analysis of the seasonal and weekly diel average variations of BTEX at 14 online monitoring sites (Figures S5 and S6) showed pattern similar to the overall diel trend. Specifically, certain monitoring sites (*i.e.*, ATH_UB (Athens), BCN_UB (Barcelona), LND_UB (London), MAR_UB (Marseille), PAR_UB (Paris-Paris 1er Les Halles), ZUR_UB (Zurich), HEL_TR (Helsinki), LND_TR (London), LYO1_IND (Lyon-Feyzin stade)) exhibited prominent morning and evening peaks during weekdays (Monday to Friday). This phenomenon is attributed to increased emissions from vehicle exhaust, as suggested by measurements in tunnels (Ammoura et al., 2014). However, during weekends (Saturday and Sunday), particularly on Sunday mornings, there were either no peaks or much weaker ones due to lower traffic density. Compared to the corresponding peaks on weekdays, the peaks on Saturday and Sunday evenings (~ 22:00-24:00) were delayed. This delay is linked to the fact that people tend to stay at home during weekends, engaging in leisure activities, and thus leading to higher emissions from cooking and household heating, and/or go out socially in the evening. These phenomena are similar to our research on levels of lung deposited surface area (LDSA) across various monitoring sites in Europe (Liu et al., 2023) and the diel variations of black carbon across different seasons in Augsburg (Liu et al., 2022).

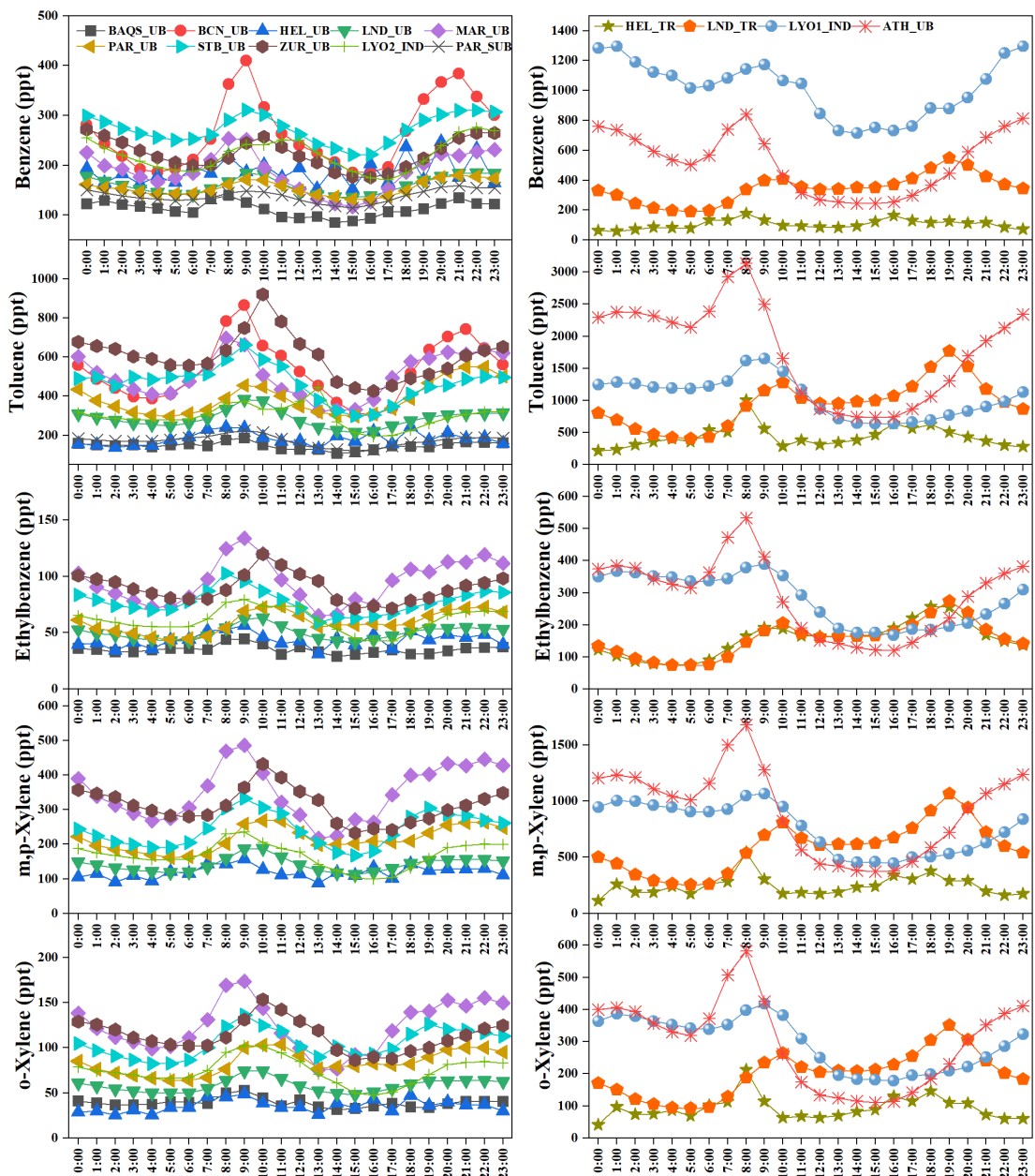

Figure 3. Diel variations of hourly average BTEX mixing ratios for 14 of the 22 studied sites, presented in local time over the entire sampling period for each site. Please note that the two vertical axis of each compound span different ranges of mixing ratios.

**3.4 Specific ratios of BTEX**

3.4.1 Benzene/Toluene (B/T)

Benzene and toluene have been found to be relatively more stable than xylenes having lifetimes of 9.5 days, 2.1 days and 7.8 h, respectively (Atkinson and Arey, 2003; Liu et al., 2008). Therefore, B/T remains relatively constant close to emission sources. However, this ratio can change over time and with the atmospheric processing of air masses (Seco et al., 2013). B/T values around 0.5 suggest significant influences from vehicle exhaust, while values significantly higher than 0.5 indicate that other sources such as industry, coal combustion, and biomass burning are major sources of BTEX species (Barletta et al., 2005; Baudic et al., 2016; Salameh et al., 2019). Additionally, higher B/T ratios can also be attributed

to the fact that emissions and/or air masses further away from the source are more aged, leading to relatively more toluene disappearance due to its higher reactivity compared to benzene.

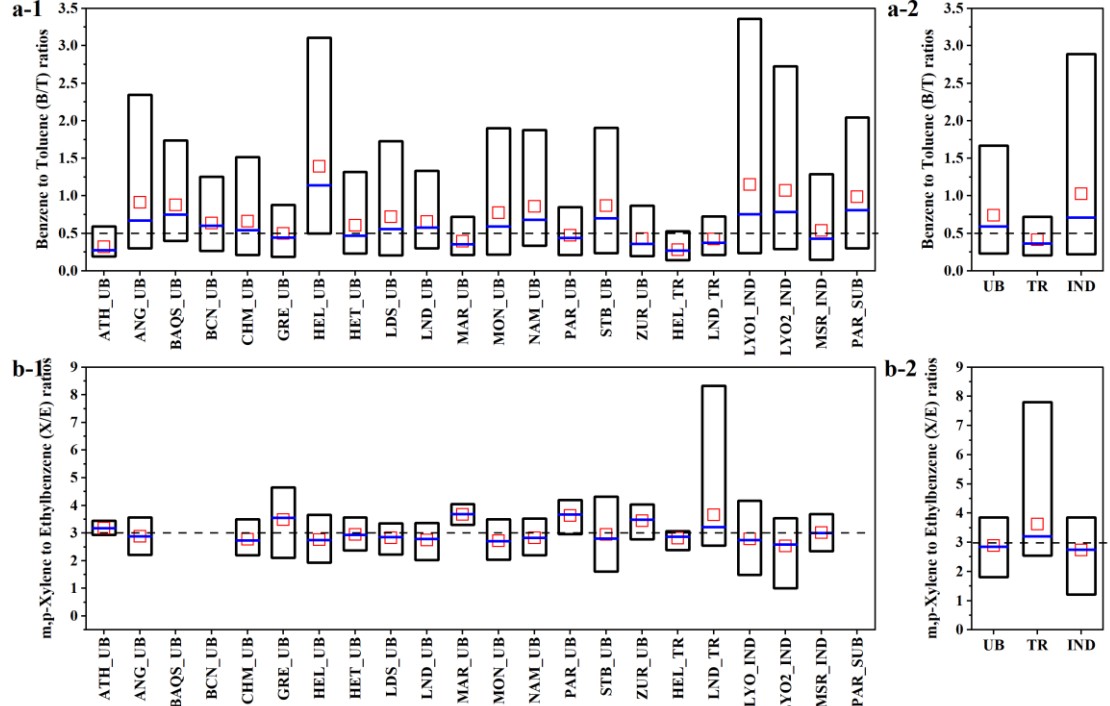

Figure 4. The ratios of benzene to toluene (B/T) and meta/para-xylene to ethylbenzene (X/E) at all sites (a-1, b-1) and different types sites (UB, urban background; TR, traffic; IND, industry; a-2, b-2). The box represented the 5th–95th percentiles of ratios. The middle line and middle square represented the median

and mean values of ratios, respectively. Blank means no available value.

As shown in Figure 4a-1, the B/T ratio ranges from $0.29 \pm 0.11$ to $1.35 \pm 0.95$, with the majority of sites averaging around 0.4-0.6 (BCN_UB (Barcelona), CHM_UB (Charleroi), GRE_UB (Grenoble), HET_UB (Herstal), MAR_UB (Marseille), PAR_UB (Paris - Paris 1[er] Les Halles), ZUR_UB (Zurich),

LND_TR (London), and MSR_IND (Mouscron)). Similarly, Salameh et al. (2019) found that at UB sites with weekly measurements, B/T was in the same order of magnitude as the TR sites with an average ratio of 0.34 in Paris. This finding was indirectly confirmed by the diel variation of BTEX mixing ratios (Figure 3) and the B/T trends in different site typologies (TR: $0.41 \pm 0.29$; UB: $0.73 \pm 0.68$; IND: $1.03 \pm 1.00$, Figure 4a-1). Notably, the B/T ratio is higher at industrial (IND) sites, which can be attributed to

the different atmospheric lifetimes of toluene and benzene. Although toluene is commonly emitted in greater quantities from industrial sources, its atmospheric lifetime is much shorter than that of benzene (toluene: 2.1 days, benzene: 9.5 days). As a result, even though toluene emissions may be significant, the shorter lifetime of toluene leads to its rapid degradation in the atmosphere compared to benzene. This allows benzene to accumulate relative to toluene, particularly near industrial sources, resulting in a higher

B/T ratio despite toluene's greater initial emissions (Atkinson and Arey, 2003; Liu et al., 2008)

In addition, based on only summer and autumn data being available (Table 1), the B/T at HEL_TR (Helsinki), serving as a traffic site, was $0.29 \pm 0.11$, suggesting vehicular exhaust emissions as the

primary source of benzene and toluene in the study area's ambient air, primarily through fuel evaporative emissions. At HEL_UB (Helsinki), the B/T ratio is relatively high, which can be attributed to the limited monitoring data available only from February 2016, introducing a potential comparability bias. For PAR_SUB (Paris-SIRTA), the elevated B/T ratio can be attributed to seasonal factors (Simon et al., 2023). Benzene concentrations typically increase from September to April, driven by more active sources during the winter months, such as residential wood burning (Languille et al., 2020), and limited dispersion due to a lower boundary layer (Simon et al., 2023). This trend is further supported by the presence of wood-burning tracers like furfural and benzenediol, which exhibit similar seasonal patterns. Although toluene does not show as strong a seasonal variation as benzene, it also has higher levels during autumn and winter. The primary source of toluene, traffic, remains important throughout the year, and more stagnant conditions in these seasons contribute to the accumulation of pollutants.

Further, considering the seasonal B/T ratios in different urban environments (Figure 5a), each monitoring site (Figure S7), and temporal trends (Figure S8) in B/T across monitoring sites, the seasonal fluctuations of benzene and toluene are consistent with B/T, showing seasonal differences to a certain extent. To analyze these variations more accurately, major axis regression (MA) was applied, accounting for uncertainties in both the x and y variables. Specifically, the mean and median values of B/T across all monitoring sites exhibit lower values in summer and higher values in winter. This pattern is consistent across different types of environments. The higher B/T values in winter further indicate influences from biomass combustion sources.

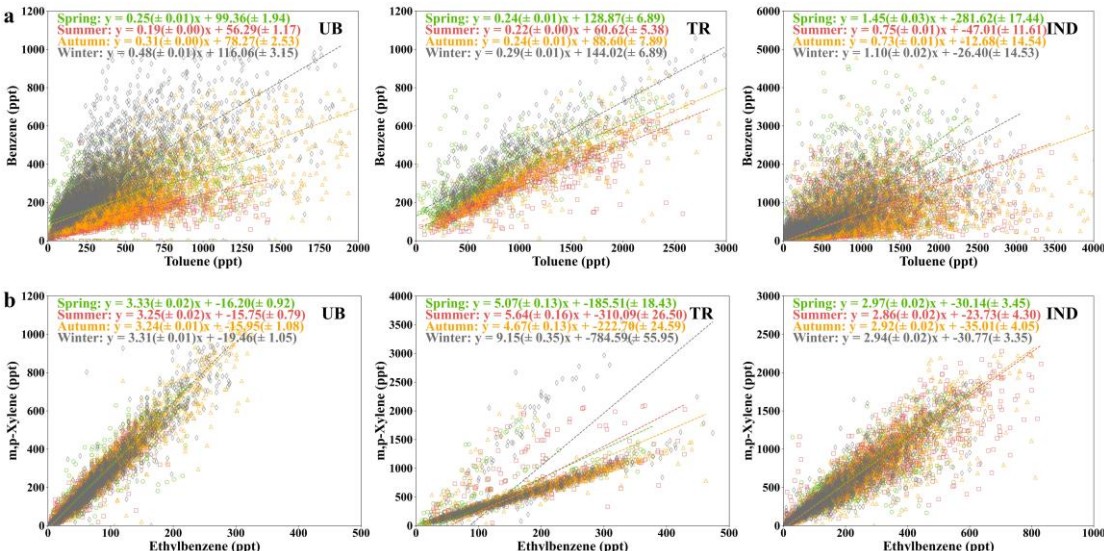

Figure 5. Regression of benzene to toluene (a) and m,p-xylene to ethylbenzene (b) ratios in different urban locations (UB, urban background; TR, traffic; IND, industry) across different seasons (green, spring; red, summer; gold, autumn; gray, winter).

3.4.2 m,p-Xylene/Ethylbenzene (X/E)

In the atmosphere, m,p-xylene and ethylbenzene are homologous and the reaction rate of m,p-xylene with the OH radical is three times faster than that of ethylbenzene (Martins et al., 2008; Han et al., 2017). Hence, the X/E ratio can serve as an indicator reflecting the extent of photochemical reactions (McKeen

and Liu, 1993), where higher and lower X/E ratios suggest local sources and external transport, respectively (Zalel and Broday, 2008).

Typical X/E ratios measured in the urban atmosphere are around 3 (Monod et al., 2001; Hsieh et al., 2011; Amodio et al., 2013). Wang et al. (2015) observed significant variations in X/E ratios (ranging from 1.2 to 2.8) in tunnels characterized by different types of vehicles in downtown Shanghai. In this study, the mean X/E ratio across monitoring sites ranged from 1.75 ± 0.91 to 3.68 ± 0.30 (Figure 4b-1), with X/E values oscillating around 3 at all sites. Additionally, there were minimal differences in the average X/E values across seasons (Figures 5b and S9) and different types of sites (Figure 4b-2). This result suggests that pollutants at the monitoring sites are primarily emitted directly from local sources (with higher X/E values). Specifically, considering the temporal variations of X/E at each monitoring site, except for the monitoring period at HEL_UB (Helsinki) in 2016, this ratio was generally greater than 2 at the other sites (Figure S7). This result further suggests that direct emissions from local sources dominate at each monitoring site. Notably, in general, the slightly lower X/E value observed at the urban background (UB) sites (*i.e.*, ANG_UB (Angleur), CHM_UB (Charleroi), GRE_UB (Grenoble), HET_UB (Helsinki), MON_UB (Mons), NAM_UB (Namur), LDS_UB (Lodelinsart), and ZUR_UB (Zurich)) in summer and the higher X/E value observed at the traffic (TR) site (*i.e.*, LND_TR (London)) in winter can be attributed to seasonal differences in photochemical reactions and atmospheric conditions, the impact of changing gasoline formulations, as well as the significant increase in BTEX concentrations during winter cold starts (George et al., 2015; Zhang et al., 2022) (Figure 5b). In summer, increased solar radiation and higher temperatures promote photochemical reactions (Rad et al., 2014), and since the reaction rate of m,p-xylene with the OH radical is three times faster than that of ethylbenzene, the X/E ratio decreases. Conversely, in winter, the formation of a stable atmospheric layer and lower temperatures may cause pollutants to remain in the atmosphere for longer periods, leading to an increased X/E ratio.

**3.5 Health risk assessment of BTEX**

Figure 6a presents LCR values associated with benzene and ethylbenzene exposure through inhalation at all monitoring sites. Across all sites, the average LCR values range from $2.6 \times 10^{-6}$ to $1.9 \times 10^{-5}$, showing that all values are below the definite risk of cancer threshold ($10^{-4}$) but exceeding the threshold established by the US (EPA, 2001) guidelines of $1 \times 10^{-6}$. This result indicates that the risk from benzene and ethylbenzene is at a moderate level and still warrants attention, especially in traffic (TR) and industry (IND) environments. It should be noted that the EPA values are based on the assumption of continuous exposure, which typically does not occur outdoors, where individuals spend less time compared to indoors. Thus, our findings apply primarily to outdoor exposure, and indoor air concentrations of BTEX compounds would need to be considered to draw comprehensive health-related conclusions.

The variability in LCR values across different sites highlights the importance of localized assessments and tailored interventions. Urban areas, for example, might need different pollution control strategies due to differences in pollutant sources and population density. Industrial and traffic-related areas often have higher concentrations of these pollutants due to emissions from vehicles and industrial activities. Therefore, populations in these areas might be at greater risk and require more stringent

monitoring and regulatory measures to mitigate exposure.

For HI, all falling below the threshold limit value (1) (Figure 6b) set by the United States Environmental Protection Agency in 2009 (EPA, 2009). This indicates a generally low non-carcinogenic risk from outdoor exposure to BTEX in the region, with levels mostly within safe thresholds. Therefore, there is no immediate risk of developing non-cancer diseases due to the inhalation of BTEX at the observed levels outdoors. However, it is important to note that long-term exposure, even within these safe limits, can still adversely affect health. Notably, sites LYO1_IND (Lyon - Feyzin stade) and LND_TR (London) exhibit higher HI values compared to others, attributed to elevated concentrations of benzene. This observation aligns with the findings reported by Jia et al. (2021) regarding VOC HI values in the Chinese Delta region and by Bretón et al. (2022) regarding BTEX HI values in the Southeast Mexico.

Furthermore, the PTR-MS systems used at BCN_UB (Barcelona) and PAR_SUB (Paris-SIRTA) were not able to distinguish the mixing ratios of ethylbenzene separate from xylenes. This limitation affects the comparability of LCR values across different monitoring sites, as the absence of ethylbenzene data at these sites results in an incomplete assessment of the total carcinogenic risk. Despite these limitations, the comparisons still offer valuable insights into the spatial distribution of cancer risks and highlight areas where more comprehensive monitoring is needed.

To enhance the reliability and comprehensiveness of future assessments, it is crucial to establish a standardized observation platform. Such a platform would ensure consistent data collection methods, pollutant measurement techniques, and risk assessment criteria across all monitoring sites. This would enable better-informed decision-making for public health interventions specifically focused on outdoor exposures.

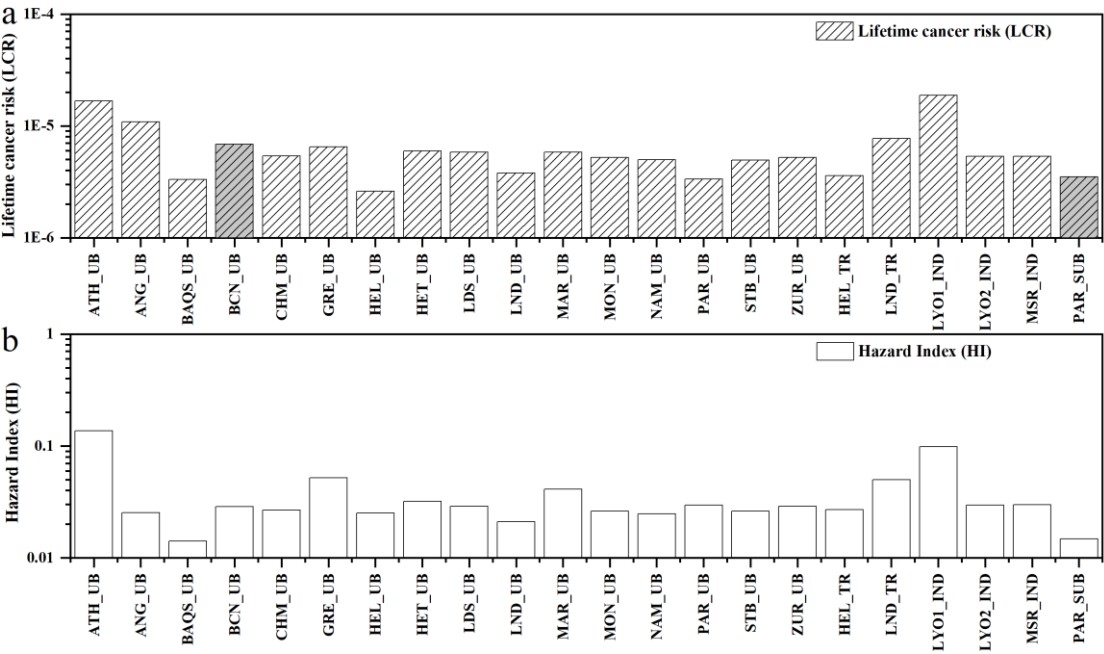

Figure 6. The lifetime cancer risk (LCR) values for the carcinogenic risk of benzene and ethylbenzene and hazard index (HI) for the carcinogenic risk of benzene, toluene, ethylbenzene, m,p-xylene, o-xylene through inhalation for individuals at all sites. Please note that the LCR values of BCN_UB and PAR_SUB have only been calculated with benzene data (marked gray)., and HI values of BAQS_UB

did not detect m,p-xylene

**4 Conclusions**

This study conducted a comprehensive assessment of long and short-term mixing ratios of benzene, toluene, ethylbenzene, m,p-xylene, o-xylene (BTEX) across 22 monitoring sites in seven European countries (Belgium, France, Finland, Greece, Spain, Switzerland, and the United Kingdom,). These sites included 3 industrial (IND) sites, 2 traffic (TR) sites, 16 urban background (UB) sites, and 1 suburban background (SUB) site. The median of BTEX mixing ratios was 772 ppt across all monitoring sites. Our findings demonstrate significant seasonal and diel variabilities in BTEX mixing ratios (p=0.01) at each monitoring site, indicating the influence of changes in traffic volume, emission sources, photo-reactivity, and meteorological factors. The B/T ratio ranged from $0.29 \pm 0.11$ to $1.35 \pm 0.95$, and the X/E ratio ranged from $1.75 \pm 0.91$ to $3.68 \pm 0.30$, demonstrating spatial variations in BTEX emission sources across monitoring sites. These results highlight that local traffic emissions are the major source of BTEX pollution, with additional contributions from industrial processes, solvents use, coatings, biomass burning, and fuel evaporation. Furthermore, the health index (HI) values of BTEX at monitoring sites were generally lower than the threshold limit value, suggesting a low non-carcinogenic risk overall. However, our health risk assessment indicates the lifetime cancer risk (LCR) from inhaling benzene and ethylbenzene, although relatively low ($<10^{-4}$), still warrants attention due to the potential effects for long-term exposure.

In conclusion, this study provides new insights into the temporal and spatial variabilities of BTEX mixing ratios across different types of sites in multiple European countries. It emphasizes the significant impact of traffic-related emissions and seasonal factors on BTEX levels, and underscores the need for continued monitoring and regulation to mitigate potential health risks. While we recognize that transportation and industrial activities are key contributors to BTEX pollution, our current data do not allow us to quantify their individual contributions. The findings also suggest that future studies should focus on the impact of changing gasoline formulations and vehicle technologies on BTEX emissions, as well as the effectiveness of current air quality regulations in reducing BTEX exposure. However, our primary aim is to provide a comprehensive assessment of the health risks associated with BTEX exposure, highlighting the urgent need for effective management strategies.

**Data availability**

The code used to generate the figures in this paper is available from the corresponding authors upon request.

**Author Contributions**

**XL:** Writing-original draft, Writing-review & editing, Conceptualization. **XZ, TW:** Methodology, Formal analysis. **LW:** Software. **MM:** Project administration. **MD, RL, RS, AMY, MG, PP, AC, MV, YF, AB, VG, GG, LL, VG, LS, HH, SR, ZB, MJM, DB, SH, ZD, RMH, WB, JD, SS, AL, JH, EL, NM:** Data curation, Writing – review & editing. **XD, TA, XZ, PKH, AA, TS:** Writing – review & editing. **XQ:** Writing – review & editing, Supervision, Project administration, Funding acquisition.

**Competing interests**

The contact author has declared that none of the authors has any competing interests.

**Acknowledgements**

This study is supported by the RI-URBANS project (Research Infrastructures Services Reinforcing Air Quality Monitoring Capacities in European Urban and Industrial Areas, European Union's Horizon 2020 research and innovation program, Green Deal, European Commission, contract 101036245). This study is also part funded by the National Natural Science Foundation of China (42407566, 42205099), the Chunhui Project Foundation of the Education Department of China (HZKY20220053), and Natural Science Foundation of Xinjiang Uygur Autonomous Region (2023D01A57). VOC data was assessed and evaluated by two ACTRIS CiGas (https://www.actris.eu/topical-centre/cigas) units at IMT Nord Europe - France and EMPA - Switzerland based on ACTRIS recommendations and guidelines. AMYS acknowledges the "Agencia Estatal de Investigación" from the Spanish Ministry of Science, Innovation and Universities for her Ramón y Cajal grant (RYC2021-032519-I) and the support from the Consolidación Investigadora project (CNS2022-135757). RS acknowledges a Ramón y Cajal grant (RYC2020-029216-I) funded by MICIU/AEI/ 10.13039/501100011033 and by "ESF Investing in your future". IDAEA-CSIC is a Severo Ochoa Centre of Research Excellence (MCIN/AEI, Project CEX2018-000794-S). NM and El acknowledge Dr. A. Panopoulou for measurements and data curation. We would also like to thank the Swiss Federal Office for the Environment (FOEN).

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
