# Peer review of "Exploring the variations in ambient BTEX in urban Europe and its environmental health implications"

_EGUsphere, 2024_

## Author Comment (AC1)

**RC1:**

The manuscript entitled " Exploring the variations in ambient BTEX in urban Europe and its environmental health implications" aimed to offer essential insights into BTEX pollution in European urban environments. However, No more new idea and deep insights about the pollution characteristics, as well as the environmental risk of BTEX pollution were put forth in the manuscript. In my opinion, the current manuscript can't be recommended for publication in Atmospheric Chemistry and Physics.

Response: Thank you for your valuable feedback on our manuscript entitled "Exploring the variations in ambient BTEX in urban Europe and its environmental health implications." We appreciate your insights and the opportunity to clarify and expand upon the contributions of our research.

Addressing Concerns Regarding Novelty and Insights

(1) Identification of Pollution Sources: Our study elucidates the distinct sources of BTEX pollution in urban environments by analyzing the mixing ratios across various site types (urban background, traffic, and industrial). The finding that traffic and industrial activities are significant contributors to BTEX pollution not only confirms existing knowledge but also adds specific spatial and temporal data, enhancing the understanding of urban pollution dynamics in Europe.

(2) Spatiotemporal Analysis: We conducted a comprehensive spatiotemporal analysis of BTEX levels across 22 European urban monitoring sites over several years (2017-2022). This provides a robust dataset that highlights not only the variability of BTEX concentrations in different urban settings but also identifies trends over time, reflecting the effectiveness of air quality regulations in some regions. Our results offer a unique perspective on how BTEX pollution levels are changing, which has implications for policy-making and public health strategies.

(3) Seasonal and Diel Variations: Our investigation into seasonal and diel variations in BTEX mixing ratios reveals important insights into how atmospheric conditions and human activities influence pollutant concentrations. This level of detail enhances the understanding of BTEX pollution's temporal dynamics, which has not been extensively studied in the context of European cities at this large scale.

(4) Environmental and Health Implications: While the manuscript outlines the characteristics of BTEX pollution, it also emphasizes the environmental and health risks associated with these compounds, particularly their role in contributing to air quality issues and potential health impacts. Additionally, we expanded the current risk assessment to include non-cancer risks associated with benzene and its derivatives. We argue that understanding these pollution characteristics is essential for evaluating public health risks and formulating effective mitigation strategies.

(5) Future Research Directions: We propose several avenues for future research, including the investigation of BTEX emissions from specific sources (e.g., residential heating, solvent use, industrial processes) and the long-term health effects of chronic exposure to BTEX in urban populations. This highlights the need for continued research in this area and underscores the relevance of our findings.

In summary, we believe that our manuscript contributes with important insights into the variations in BTEX pollution across urban Europe and its environmental health implications. We appreciate your suggestion for a deeper exploration of the pollution characteristics, and we will take this

opportunity to revise the manuscript to better articulate these insights and their implications.

Specific comments:

1. The first is that the method is not innovative (only analyzing spatio-temporal changes, characteristic ratios, and health risks).

Response: Thank you for your valuable comment. We acknowledge the reviewer's concern regarding the lack of methodological innovation in our analysis. While the methods employed—spatio-temporal changes, characteristic ratios, and health risk assessments—are established techniques, they provide a comprehensive framework for understanding BTEX pollution dynamics in urban Europe. Our aim was to synthesize existing knowledge and present a thorough analysis of BTEX pollution characteristics across diverse urban environments. By focusing on European urban areas, we contribute to the existing literature by contextualizing these methods within a specific geographical framework and highlighting trends and variations that may inform future studies and policies.

2. The second is that the conclusion is also not innovative (the main source is not quantitative, many of which are the results of previous research).

Response: We appreciate your feedback regarding the perceived lack of innovation in our conclusions. We have revised the conclusion to emphasize the quantitative insights gained from our analysis and how they contribute to the existing body of knowledge on BTEX pollution. Specifically, we have clarified that, while we recognize transportation and industrial activities as significant contributors to BTEX pollution, our current data do not allow us to separately quantify their individual contributions. Additionally, we emphasize that the primary aim of our study was to assess the health impact based on quantified indicators, as indicated in the title, rather than to conduct a detailed source apportionment analysis of BTEX. Our approach focuses on providing a comprehensive assessment of the health risks associated with BTEX exposure, while acknowledging the limitations in quantitatively distinguishing the contributions from various sources.

The revised text can be found in the conclusion section (Lines 490-492):

"While we recognize that transportation and industrial activities are key contributors to BTEX pollution, our current data do not allow us to quantify their individual contributions."

We hope this revision addresses your concerns and strengthens the manuscript's contribution to the understanding of BTEX pollution.

3. Thirdly, the results were relatively arbitrary and did not take into account the influence of various factors on the external field. For example, in the section of Line 182-187, it is not very appropriate to conclude that transportation and industrial activities are the main sources of BTEX pollutants in the study area based solely on the different concentrations of different types of pollutants, as the meteorological conditions in these locations may vary greatly.

Response: We appreciate the reviewer's comments regarding the need to account for various factors that can influence BTEX concentrations and sources. We acknowledge that meteorological conditions and other external factors play a significant role in determining BTEX levels and should not be overlooked. In our manuscript (line 196-211), we initially provided a preliminary analysis based on available data to identify potential sources. To strengthen our conclusions, we applied a commonly used method to determine urban enhancement ratios (ER) (Salameh et al., 2019, https://doi.org/10.1016/j.aeaoa.2018.100003), estimating the slopes of least-square linear regressions between each TEX compound and benzene. By using ER, we reduce the sensitivity of

the analysis to background conditions, dilution, and air-mass mixing compared to using absolute concentrations (Salameh et al., 2019).

Our results show spatial differences in the ER values for TEX/B, with the highest ratios observed at TR sites, followed by UB sites, and the lowest at IND sites. Specifically, the slopes were 2.09±0.05 for T/B, 0.37±0.01 for E/B, 1.21±0.03 for m,p-X/B, and 0.48±0.01 for o-X/B at TR sites, 1.57±0.02 for T/B, 0.23±0.00 for E/B, 0.71±0.01 for m,p-X/B, and 0.27±0.00 for o-X/B at UB sites, and 0.37±0.01 for T/B, 0.13±0.00 for E/B, 0.29±0.01 for m,p-X/B, and 0.13±0.00 for o-X/B at IND sites. A similar trend was observed in the seasonal variations, with ER values generally following the order TR > UB > IND. Notably, for UB and TR sites, the ER for TEX/B was higher in summer, while for IND sites, the ER was lowest during summer (Table S1). These findings suggest that the additional evaporative sources, potentially related to traffic or solvent usage, particularly at urban background sites, may contribute to the observed seasonal and spatial variations.

Additionally, our study examined diurnal variations, which showed that meteorological and photochemical processes (e.g., daytime vs. nighttime conditions) also influence BTEX levels, supporting the complex interplay of various factors impacting the pollutant concentrations.

*Table S1. Urban enhancement ratios (ER) of different types (urban background, UB; traffic, TR; industrial site, IND) in different seasons.*

| Seasons | Types | Toluene vs. Benzene | Ethylbenzene vs. Benzene | m,p-Xylene vs. Benzene | o-Xylene vs. Benzene |
|---------|-------|---------------------|--------------------------|------------------------|----------------------|
| Spring | UB | 1.69 ± 0.05 | 0.23 ± 0.01 | 0.67 ± 0.05 | 0.3 ± 0.02 |
| Spring | TR | 2.19 ± 0.09 | 0.36 ± 0.01 | 1.13 ± 0.05 | 0.45 ± 0.02 |
| Spring | IND | 0.46 ± 0.02 | 0.13 ± 0.01 | 0.30 ± 0.01 | 0.13 ± 0.01 |
| Summer | UB | 2.64 ± 0.06 | 0.32 ± 0.01 | 1.02 ± 0.05 | 0.37 ± 0.02 |
| Summer | TR | 3.28 ± 0.20 | 0.55 ± 0.02 | 1.82 ± 0.07 | 0.69 ± 0.03 |
| Summer | IND | 0.30 ± 0.01 | 0.10 ± 0.01 | 0.21 ± 0.01 | 0.09 ± 0.01 |
| Autumn | UB | 1.70 ± 0.04 | 0.20 ± 0.01 | 0.60 ± 0.01 | 0.25 ± 0.01 |
| Autumn | TR | 2.54 ± 0.06 | 0.49 ± 0.02 | 1.65 ± 0.06 | 0.65 ± 0.02 |
| Autumn | IND | 0.42 ± 0.03 | 0.24 ± 0.01 | 0.49 ± 0.02 | 0.23 ± 0.01 |
| Winter | UB | 1.76 ± 0.03 | 0.29 ± 0.01 | 0.88 ± 0.02 | 0.32 ± 0.01 |
| Winter | TR | 1.98 ± 0.05 | 0.33 ± 0.01 | 1.04 ± 0.05 | 0.42 ± 0.02 |
| Winter | IND | 0.38 ± 0.02 | 0.10 ± 0.01 | 0.27 ± 0.01 | 0.12 ± 0.01 |

---

## Author Comment (AC2)

**RC2:**

The manuscript, while providing and overview over several BTEX datasets in one place to me appears to be lacking depth in the analysis and truly novel results. To me it appears that the paper is more of a measurement report paper. Hence while I am not convinced about novel findings related to atmospheric chemistry, I do believe it can be considered as a measurement report.

However, I have several concerns regarding the statistical analysis presented:

It is clear from the box plots in Figures in the supplement and main text that the data does not follow a normal distribution at any of the sites. The mean is always significantly higher than the median. It is very likely that all these distributions will fail a normality test when subjected to one hence statistical parameters used to represent the data must be robust parameters.

My statistics and data analysis related concerns are as follows:

1)Average and standard deviation are not the correct statistical parameters to report when data fails the normality test. Instead, the median would be more appropriate than the mean. Sigma can be approximated in a robust manner from the median median deviation MAD (1 sigma = 1.48*MAD). Authors can also report the median and the range of values observed instead.

Response: We acknowledge that reporting the mean and standard deviation may not be appropriate given the non-normal distribution of our data. In the revised manuscript, we have replaced these metrics with the median and the median absolute deviation (MAD) as robust statistical measures. We have added the following sentence to the method section (line 182-184):

Therefore, in this study, BTEX levels are presented as median mixing ratios (MED) ± median absolute deviation (MAD) in parts per trillion by volume (ppt) to reflect the non-normal distribution of the data.

2) The difference of median between seasons must be assessed with a robust test such as Mann-Whitney test since the data is non-normal. The robust substitute of ANOVA would be a Kruskal-Wallis test for equal median although a Mann-Whitney pairwise comparison between season may turn out to be more revealing and interesting.

Response: Thank you for your suggestion. We agree that a robust test is necessary given the non-normality of the data. Therefore, we have replaced the current analysis with the Mann-Whitney test to assess the differences in BTEX levels between seasons. Additionally, we incorporated a Kruskal-Wallis test for assessing equal medians and conducted pairwise Mann-Whitney comparisons between seasons, as suggested. The results still indicate significant differences in BTEX levels across the seasons. Although the conclusions are consistent with those obtained using the previous ANOVA approach, we followed the reviewer's recommendation to apply the Mann-Whitney test to ensure a more robust analysis of the non-normally distributed data.

**Test Statistics[a]**

|  | Benzene | Toluene | Ethylbenzene | m,p-Xylene | o- Xylene |
|---|---|---|---|---|---|
| Mann-Whitney U | 11141561.500 | 22785960.500 | 19782083.000 | 19342025.000 | 20020547.500 |
| Wilcoxon W | 24333377.500 | 36501663.500 | 33211236.000 | 32349575.000 | 33387582.500 |
| Z | -55.716 | -13.506 | -22.179 | -21.629 | -20.893 |
| Asymp. Sig. (2-tailed) | 0.000 | 0.000 | 0.000 | 0.000 | 0.000 |
| a. Grouping Variable: Spring-Summer | | | | | |

**Test Statistics[a]**

|  | Benzene | Toluene | Ethylbenzene | m,p-Xylene | o- Xylene |
|---|---|---|---|---|---|
| Mann-Whitney U | 41042737.000 | 37480442.500 | 35590465.000 | 34944741.000 | 35669694.500 |
| Wilcoxon W | 88510633.000 | 87846108.500 | 83497831.000 | 81762067.000 | 83420572.500 |

| | | | | | |
|---|---|---|---|---|---|
| Z | -14.329 | -27.996 | -27.245 | -26.466 | -26.513 |
| Asymp. Sig. (2-tailed) | 0.000 | 0.000 | 0.000 | 0.000 | 0.000 |
| a. Grouping Variable: Spring-Autumn | | | | | |

**Test Statistics[a]**

| | Benzene | Toluene | Ethylbenzene | m,p-Xylene | o- Xylene |
|---|---|---|---|---|---|
| Mann-Whitney U | 29642393.000 | 41652262.500 | 39365751.500 | 39224692.500 | 39342415.000 |
| Wilcoxon W | 77110289.000 | 92017928.500 | 87273117.500 | 86042018.500 | 87093293.000 |
| Z | -43.891 | -17.123 | -17.238 | -15.662 | -16.540 |
| Asymp. Sig. (2-tailed) | 0.000 | 0.000 | 0.000 | 0.000 | 0.000 |
| a. Grouping Variable: Spring-Winter | | | | | |

**Test Statistics[a]**

| | Benzene | Toluene | Ethylbenzene | m,p-Xylene | o- Xylene |
|---|---|---|---|---|---|
| Mann-Whitney U | 9224353.000 | 15511278.000 | 12996645.000 | 12702955.500 | 13457629.000 |
| Wilcoxon W | 22416169.000 | 29226981.000 | 26425798.000 | 25710505.500 | 26824664.000 |
| Z | -62.518 | -39.336 | -46.748 | -46.051 | -44.463 |
| Asymp. Sig. (2-tailed) | 0.000 | 0.000 | 0.000 | 0.000 | 0.000 |
| a. Grouping Variable: Summer-Autumn | | | | | |

**Test Statistics[a]**

| | Benzene | Toluene | Ethylbenzene | m,p-Xylene | o- Xylene |
|---|---|---|---|---|---|
| Mann-Whitney U | 4764916.000 | 17959649.000 | 15280818.000 | 15304223.000 | 15663183.000 |
| Wilcoxon W | 17956732.000 | 31675352.000 | 28709971.000 | 28311773.000 | 29030218.000 |
| Z | -80.742 | -29.284 | -37.256 | -35.495 | -35.121 |
| Asymp. Sig. (2-tailed) | 0.000 | 0.000 | 0.000 | 0.000 | 0.000 |
| a. Grouping Variable: Summer-Winter | | | | | |

**Test Statistics[a]**

| | Benzene | Toluene | Ethylbenzene | m,p-Xylene | o- Xylene |
|---|---|---|---|---|---|
| Mann-Whitney U | 34862699.500 | 42758108.000 | 40750515.000 | 39671945.000 | 40395430.000 |
| Wilcoxon W | 80602529.500 | 89440061.000 | 84878830.000 | 83219723.000 | 84045926.000 |
| Z | -28.606 | -10.573 | -9.253 | -9.948 | -9.198 |
| Asymp. Sig. (2-tailed) | 0.000 | 0.000 | 0.000 | 0.000 | 0.000 |
| a. Grouping Variable: Autumn-Winter | | | | | |

**Test Statistics[a,b]**

| | Benzene | Toluene | Ethylbenzene | m,p-Xylene | o- Xylene |
|---|---|---|---|---|---|
| Kruskal-Wallis H | 7514.139 | 1791.224 | 2398.635 | 2256.100 | 2181.433 |
| df | 3 | 3 | 3 | 3 | 3 |
| Asymp. Sig. | 0.000 | 0.000 | 0.000 | 0.000 | 0.000 |
| a. Kruskal Wallis Test | | | | | |
| b. Grouping Variable: Seasons | | | | | |

3) All the fits in Figure 5 appear to be ordinary least square regressions despite the fact that both x and y axis contain measured data that carries measurement uncertainties. OLR is only suitable when the quantity on the X-Axis is absolute and error free. Please use major axis regression (MA) also

known as orthogonal regression to fit your data. Also: You need to give the error of the slope and intercepts. Figure 5B urban panel the line fit for summer doesn't appear to have much to do with the data at least I can't see the corresponding circles anywhere near the line. The slope may be driven by some extreme values outside the x and y axis boundaries. Please check your data

Response: Thank you for your insightful comments. We appreciate your input regarding the use of ordinary least squares regression (OLR) and its suitability for our data. In response, we have employed major axis regression (orthogonal regression) for our analyses, which better accounts for measurement uncertainties in both x and y variables. We have also included the error estimates for the slope and intercept in the revised Figure 5. We have updated the Figure 5 to reflect this change. Furthermore, we have re-evaluated the data points in Figure 5B and ensured that the regression line corresponds appropriately with the observed data points, removing any misleading extremes.

[Figure]

*Figure 5. Regression of benzene to toluene (a) and m,p-xylene to ethylbenzene (b) ratios in different urban locations (UB, urban background; TR, traffic; IND, industry) across different seasons (green, spring; red, summer; gold, autumn; gray, winter).*

My other concerns are es follows

1) Referring to sites with the site code instead of the name of the city makes the manuscript hard to follow. It would be nicer to refer to them with the name of the city and nature of the site even if it costs a few more words.

Response: Thank you for your comments. Regarding the concern about referring to the sites with codes instead of city names: The cities mentioned in the manuscript are part of the RI-URBANS project, and the site codes (i.e., the first three capitalized letters of the city name followed by the site type: UB, TR, IN) are standardized across all related documentation. This format ensures consistency throughout the project and allows for easy reference to the sites across different studies. Therefore, while we understand that using city names might make the text more intuitive for some readers, we believe that adhering to this standard is essential for maintaining uniformity within the context of the broader project. To assist readers, we have included Table 1, which provides a clear reference linking the site codes to the corresponding cities and site types, thereby helping to clarify

the manuscript.

2) Please make all comparison qualitative: E.g. a statement like

"Nonetheless, compared to the period before 2000, the levels of benzene and other BTEX compounds have shown a decreasing trend due to the successful implementation of air quality measures in Greece, such as the extension of metro lines and the use of catalytic converters in cars" Implies a time series trend analysis was done which I can't find in the results section. So it appears to be a comparison between values reported in this study and a previous study. Then why not state clearly what the was then, and is now.

Response: Thank you for your valuable suggestion. Following your suggestion, we have now added the specific values to clearly indicate the comparison between the results from this study and previous studies.

"Nonetheless, compared to the period before 2000, the levels of benzene and other BTEX compounds have shown a decreasing trend due to the successful implementation of air quality measures in Greece, such as the extension of metro lines and the use of catalytic converters in cars (Panopoulou et al., 2021). For instance, benzene levels at traffic sites have decreased significantly, dropping by as much as eightfold, from approximately 12520 ppt in 1994 to about 1565 ppt in 2016 (Panopoulou et al., 2021). Similarly, at urban monitoring stations, benzene concentrations fell sharply, from around 4695 ppt during the period of 1993–1996 to between 313 and 1565 ppt in 2016 (Panopoulou et al., 2021)."

3) Please highlight and explain interesting data instead of only focusing on the data that meats the expectations of BTEX=traffic & Industry. Why is the B/T rations in urban Helsinki and suburban Paris so high?

Response: Thank you for your insightful comments. We acknowledge that the B/T ratios in urban Helsinki and suburban Paris warrant further discussion. In Helsinki (HEL_UB), the monitoring data is limited to February 2016, which may lead to a comparability bias and explain the relatively high B/T ratio observed. We have emphasized this limitation in the revised manuscript. For suburban Paris (PAR_SUB), the elevated B/T ratio can be attributed to seasonal factors. Benzene concentrations typically increase from September to April, driven by more active sources during the winter months, such as residential wood burning, and limited dispersion due to a lower boundary layer. This trend is further supported by the presence of wood-burning tracers like furfural and benzenediol, which exhibit similar seasonal patterns (https://doi.org/10.5194/essd-15-1947-2023). Although toluene does not show as strong a seasonal variation as benzene, it also has higher levels during autumn and winter. The primary source of toluene, traffic, remains important throughout the year, and more stagnant conditions in these seasons contribute to the accumulation of pollutants. We have revised the manuscript to include these explanations for a clearer understanding of the data.

Specific modifications can be found in Lines 347-350: At HEL_UB, the B/T ratio is relatively high, which can be attributed to the limited monitoring data available only from February 2016, introducing a potential comparability bias. For PAR_SUB, the elevated B/T ratio can be attributed to seasonal factors (Simon et al., 2023). Benzene concentrations typically increase from September to April, driven by more active sources during the winter months, such as residential wood burning (Languille et al., 2020), and limited dispersion due to a lower boundary layer (Simon et al., 2023). This trend is further supported by

the presence of wood-burning tracers like furfural and benzenediol, which exhibit similar seasonal patterns. Although toluene does not show as strong a seasonal variation as benzene, it also has higher levels during autumn and winter. The primary source of toluene, traffic, remains important throughout the year, and more stagnant conditions in these seasons contribute to the accumulation of pollutants.

4)Why are B/T ratio's higher at industrial sites. While toluene is known to be an industrial solvent significant toluene emission should actually lower the ratio. Industrial benzene use has supposedly been phased out due to the carcinogenic nature. Can the authors comment on the industrial benzene source? Alternatively, is this actually representative of industrial sites in Europe in general since 2/3 industrial sites are in Lyon. This could be related to local emissions from a specific industry.

Response: Thank you for your valuable comments on the B/T ratio observed at industrial sites. Although toluene is commonly emitted in greater quantities from industrial sources, its atmospheric lifetime is much shorter than that of benzene (toluene: 2.1 days, benzene: 9.5 days). As a result, even though there may be significant toluene emissions, the shorter lifetime means that it degrades more rapidly in the atmosphere compared to benzene. This leads to an accumulation of benzene relative to toluene, particularly near industrial sources, which results in a higher B/T ratio despite toluene's greater initial emissions (Atkinson and Arey, 2003; Liu et al., 2008). We believe this factor, combined with potential localized sources of benzene, could explain the relatively higher B/T ratios observed at some industrial sites.

In addition, the observed B/T ratios may not be fully representative of industrial sites across Europe, as the industrial emissions are highly dependent on specific processes at each location. In this study, two-thirds of the industrial monitoring sites are in Lyon, which may reflect local industry characteristics rather than a broader pattern across European industrial sites. Emissions can also be episodic, influenced by particular operational activities, and may vary with changes in wind direction, further complicating comparisons across regions. These factors suggest that the higher B/T ratios observed may be more indicative of localized conditions rather than a general trend for European industrial areas.

Minor corrections

Line 109-113 BTEX became TEX please correct

Response: Thank you for pointing this out. We have reviewed the original text and confirm that "TEX" is correct in this context, as the proportion of benzene in traffic emissions is relatively small compared to other TEX compounds. Therefore, the use of TEX, instead of BTEX, accurately reflects the findings of the referenced study.

---

## Author Response (AR2)

Reviewer comments:

I suggest acceptance after minor revisions regarding the following points:

- While it is understandable that the site codes are necessary and standardized, the use of solely the codes in the text makes the manuscript hard to read. Please include the site codes and city names in the text (e.g. "…with the exception of HEL_UB (Helsinki)…". Similarly, when "TR" or "IND" are used on their own, like in line 436, please add the full word for sake of readability.

**Response:** Thank you for this valuable suggestion. We agree that including both the site codes and city names will enhance the readability of the manuscript. We will revise the text accordingly by adding the city names alongside the site codes (e.g., "HEL_UB (Helsinki)") throughout the manuscript. Similarly, we will ensure that abbreviations such as "TR" and "IND" are accompanied by their full terms (e.g., "Traffic" and "Industry") where they first appear to improve clarity for the readers.

- Section 2.2 (risk assessment): the values provided by the US EPA are being used here on a European dataset, which makes the reader wonder if there are no such standards (or limits) in the EU. For example, the EU has defined OEL limits. A discussion of why EPA values are used instead is missing (or instead a comparison between the two).

**Response:** Thank you for the reviewer's insightful comment. We would like to clarify our rationale for using U.S. EPA values and provide a comparison with the European Union's Occupational Exposure Limits (OEL).

In our study, we employed U.S. EPA values because the EPA's health risk assessment framework is globally recognized for its systematic and standardized approach to evaluating environmental health risks. The Lifetime Cancer Risk (LCR) values and Reference Concentration (RfC) values provided by the EPA are widely used in assessing health risks related to air pollutants. These values are based on extensive data and evaluations of human health, making them highly reliable for evaluating risk from environmental exposures.

The EU's Occupational Exposure Limits (OEL) are important regulatory standards focused primarily on limiting acute exposure risks in occupational settings. However, these standards are designed for workplace environments rather than long-term, low-level exposure encountered in ambient environments by the general population. Consequently, OEL values are not entirely suitable for assessing chronic exposure risks among the general public. In contrast, the EPA's LCR and RfC values are specifically intended to address the health risks associated with prolonged exposure to environmental pollutants, aligning more closely with the aims of our study.

Our choice to use EPA standards does not imply a dismissal of EU standards but rather reflects the need for values that align with the specific objectives of our study. To provide further clarity, we have added a discussion in the manuscript explaining the differences between the EPA and OEL standards (line 167-173).

"*Specially, in this study, we adopted U.S. EPA values as they provide a standardized framework specifically designed for environmental exposures (Phillips and Moya, 2013). The EPA's health risk assessment values (LCR and RfC) are widely utilized due to their systematic approach and extensive data basis for chronic exposure scenarios. While the EU's Occupational Exposure Limits (OEL) are established for workplace environments, they primarily address acute exposure risks in occupational settings (Högberg and Järnberg, 2023). Therefore, they are not entirely suitable for assessing chronic, low-level exposure risks among the general public.*"

- Section 3.5 (health risk assessment): it should be mentioned that the EPA values used are based on the assumption of permanent exposure, which does not mostly happen outdoors. Most humans spend most of their time indoors, so that the contribution of outdoor air pollution does not necessarily determine health or cancer risks. This caveat (i.e. that indoor air concentrations of the BTEX compounds would be needed to make any health-related conclusions) should be discussed. Also, when mentioning health risks in the conclusions, please make clear that this is valid for outdoor exposure only. (E.g. "..the health index values of BTEX at monitoring sites were generally lower than the threshold limit value, suggesting a low non-carcinogenic risk from outdoor exposure to BTEX"). Similar adjustment is necessary when discussing cancer risk.

**Response:** Thank you to the reviewer for the thorough review of this section. We agree that the EPA's risk assessment method is indeed based on the assumption of continuous exposure, which does not entirely apply to outdoor environments. Most people spend more time indoors, so outdoor air pollution alone does not fully determine health or cancer risks. The indoor concentrations of BTEX compounds are also crucial for evaluating health-related conclusions. In light of this, we have clearly indicated in the revised manuscript that this assessment is specific to outdoor exposure and emphasized in the conclusion that our results pertain to outdoor environments.

Line 460-464

*It should be noted that the EPA values are based on the assumption of continuous exposure, which typically does not occur outdoors, where individuals spend less time compared to indoors. Thus, our findings apply primarily to outdoor exposure, and indoor air concentrations of BTEX compounds would need to be considered to draw comprehensive health-related conclusions.*

Line 472-475

*This indicates a generally low non-carcinogenic risk from outdoor exposure to BTEX in the region, with levels mostly within safe thresholds. Therefore, there is no immediate risk of developing non-cancer diseases due to the inhalation of BTEX at the observed levels outdoors.*

Line 489-490

*This would enable better-informed decision-making for public health interventions specifically focused on outdoor exposures.*

- You responded to the referee comment "Why are B/T ratio's higher at industrial sites" but did not make changes to the text that would provide that same information to the reader, at least I did not find it. (Recommendation for future revisions: it is helpful to color any text changes in the response document in a different color, so that it is easy to find out what changes to the text have been made in response to which comment.)

**Response:** Thank you for the helpful suggestion. We have marked the changes in the revised manuscript in blue to make it easier to locate updates corresponding to each comment. Specifically, the response to the B/T ratio question can be found on line 393-395 in the revised manuscript.

*"Notably, the B/T ratio is higher at industrial (IND) sites, which can be attributed to the different atmospheric lifetimes of toluene and benzene. Although toluene is commonly emitted in greater quantities from industrial sources, its atmospheric lifetime is much shorter than that of benzene (toluene: 2.1 days, benzene: 9.5 days). As a result, even though toluene emissions may be significant, the shorter lifetime of*

*toluene leads to its rapid degradation in the atmosphere compared to benzene. This allows benzene to accumulate relative to toluene, particularly near industrial sources, resulting in a higher B/T ratio despite toluene's greater initial emissions (Atkinson and Arey, 2003; Liu et al., 2008)"*

Based on the concerns of the reviewers regarding the novelty of the analysis, I suggest that the category of the manuscript will be changed from "Research Article" to "Measurement Report". Nevertheless, the manuscript provides an important dataset with comparison of BTEX concentrations between many European locations and is as such a relevant addition to the literature.

**Response:** Thank you for your thoughtful suggestion. We agree with your recommendation and will change the manuscript category from "Research Article" to "Measurement Report" as per your advice. We believe this change better reflects the nature of the manuscript, which provides valuable data and a comparison of BTEX concentrations across multiple European locations. We appreciate your input and are confident that the manuscript will be a relevant contribution to the existing literature.